# Selective deforestation and exposure of African wildlife to bat-borne viruses
Pawel Fedurek[1,2,13], Caroline Asiimwe[2,13], Gregory K. Rice[3,4], Walter J. Akankwasa[2], Vernon Reynolds[2,5], Catherine Hobaiter [2,6], Robert Kityo[7], Geoffrey Muhanguzi[2], Klaus Zuberbühler[2,6,8], Catherine Crockford[9,10], Regina Z. Cer[3], Andrew J. Bennett[3,4], Jessica M. Rothman[11], Kimberly A. Bishop-Lilly [3] & Tony L. Goldberg [12] ✉

Proposed mechanisms of zoonotic virus spillover often posit that wildlife transmission and amplification precede human outbreaks. Between 2006 and 2012, the palm *Raphia farinifera*, a rich source of dietary minerals for wildlife, was nearly extirpated from Budongo Forest, Uganda. Since then, chimpanzees, black-and-white colobus, and red duiker were observed feeding on bat guano, a behavior not previously observed. Here we show that guano consumption may be a response to dietary mineral scarcity and may expose wildlife to bat-borne viruses. Videos from 2017–2019 recorded 839 instances of guano consumption by the aforementioned species. Nutritional analysis of the guano revealed high concentrations of sodium, potassium, magnesium and phosphorus. Metagenomic analyses of the guano identified 27 eukaryotic viruses, including a novel betacoronavirus. Our findings illustrate how "upstream" drivers such as socioeconomics and resource extraction can initiate elaborate chains of causation, ultimately increasing virus spillover risk.

Spillover of viruses from wildlife to humans is often thought to be preceded by viral transmission and amplification among wildlife. For example, human ebolavirus outbreaks in Africa follow sylvatic transmission cycles in non-human primates and ungulates, with humans likely becoming infected through contact with carcasses[1–3]. Similarly, epidemiological data and analyses of inferred viral genomic recombination suggest that approximately half of human-infecting coronaviruses underwent transmission from wildlife reservoirs to humans through intermediary hosts[4,5]. Despite the high social and economic costs of zoonoses[6], the mechanisms underlying such antecedent virus transmission within animals remain poorly understood.

Budongo Forest Reserve, western Uganda, contains approximately 482 km$^2$ of medium-altitude, semi-deciduous forest[7] and is located in the Albertine Rift, a region of exceptional biodiversity and endemism[8]. Until approximately 2008, the swamp forests of Budongo contained *Raphia*

*farinifera*, a palm that, when decaying, provided a high-quality source of essential dietary minerals to wildlife[9]. Between 2006 and 2012, tobacco farming increased markedly in the area due to rising international demand and incentives from tobacco companies with longstanding operations in Uganda[10]. As a result, local farmers nearly extirpated *R. farinifera* because of its usefulness for making strings on which to dry tobacco leaves[9,11]. Budongo's eastern chimpanzees (*Pan troglodytes schweinfurthii*) altered their feeding behavior in response to this loss of a primary source of dietary minerals, increasingly consuming alternative sources such as clay, termite mounds, and the decaying pith of other tree species[9]. In 2017, we observed a never-before documented behavior by several species of wildlife in Budongo, including chimpanzees: the consumption of bat guano.

Here we present the results of an investigation as to whether this behavior could be an adaptation to dietary mineral scarcity, analogous to what has been documented for similar behaviors at this site[9]. We also

[1]Division of Psychology, Faculty of Natural Sciences, University of Stirling, Stirling FK9 4LA, UK. [2]Budongo Conservation Field Station, PO Box 362, Masindi, Uganda. [3]Biological Defense Research Directorate, Naval Medical Research Command, Fort Detrick, MD 21702, USA. [4]Leidos, 1750 Presidents St, Reston, VA 20190, USA. [5]School of Anthropology, University of Oxford, 51/53 Banbury Road, Oxford OX2 6PE, UK. [6]School of Psychology and Neuroscience, University of St Andrews; St Mary's Quad, South Street, St Andrews KY16 9JP, UK. [7]Department of Zoology, Entomology & Fisheries Sciences, Makerere University, PO Box 7062, Kampala, Uganda. [8]Institute of Biology, University of Neuchâtel, Rue Emile-Argand 11, CH-2000, Neuchâtel, Switzerland. [9]Max Planck Institute for Evolutionary Anthropology, Deutscher Platz 6, 04103 Leipzig, Germany. [10]Institut des Sciences Cognitives, 67 Bd Pinel, 69500 Bron, France. [11]Department of Anthropology, Hunter College of the City University of New York, 695 Park Avenue, New York, NY 10065, USA. [12]School of Veterinary Medicine, Department of Pathobiological Sciences, University of Wisconsin-Madison, 1656 Linden Drive, Madison, WI, USA. [13]These authors contributed equally: Pawel Fedurek, Caroline Asiimwe. ✉e-mail: tony.goldberg@wisc.edu

investigate whether guano consumption could be an ecological mechanism whereby wildlife such as chimpanzees might be exposed to bat-borne viruses. We document high frequencies of guano consumption by three species of wildlife in Budongo, high concentrations of essential dietary minerals in the guano, and diverse bat-borne viruses in the guano, including a novel betacoronavirus within the *Hibecovirus* subgenus. These results illustrate how remote upstream forces can induce unanticipated causal chains that alter wildlife ecology and behavior, one result of which may be to increase virus spillover risk.

## Results

### Field studies
On June 25, 2017, we first observed chimpanzees consuming bat guano from under a large, hollow tree (*Mildbraediodendron excelsum*) in which a colony of Noack's roundleaf bat (*Hipposideros ruber*) was roosting (Fig. 1). Using a trail camera, we captured video images of chimpanzees, black-and-white colobus (*Colobus guereza occidentalis*) and red duiker (*Cephalophus natalensis*) repeatedly consuming guano from beneath the tree (Fig. 1). Animals consumed the guano directly, and not incidentally (e.g. from consumption of adjacent contaminated clay and water), as evidenced by the clearly visible selection of the guano itself during all instances and the presence of excavations and characteristic hand prints in the guano after animals had left (Supplementary Fig. 1a). We recorded 92 separate instances of guano consumption by chimpanzees on 71 different days, with between 1 and 13 chimpanzees per instance. Chimpanzees removed and ate guano with their hands (Fig. 1b, Supplementary Videos 1 and 2), or they drank adjacent water using a leaf sponge (folded leaves used to collect water[12]; Supplementary Video 3). Cameras captured black-and-white colobus feeding on guano on 65 occasions during 56 different days, with between 1 and 9 individuals per instance. These primates ate guano directly (Fig. 1c; Supplementary Video 4). Cameras captured solitary red duikers on 682 occasions on 210 different days. Duikers either licked guano directly or drank adjacent water next to the pile (Fig. 1d; Supplementary Video 5). On one occasion, we observed a ~ 2 m human-modified pole, suggesting that local people had also visited this tree, perhaps to collect guano (Supplementary Fig. 1b).

### Dietary mineral analyses
Nutritional analyses revealed that the guano contained concentrations of magnesium, phosphorus and potassium higher than in any other recorded dietary source of minerals at Budongo (Table 1). The guano also contained concentrations of sodium approximately equal to that of decaying *Cleistopholis patens* wood, the primary alternative source of dietary sodium for chimpanzees subsequent to the loss of *R. farinifera*[9] (Table 1). Concentrations of calcium, manganese and iron were within ranges of other sources at Budongo (Table 1).

### Virus identification and characterization
Metagenomic analyses of the bat guano revealed 27 novel putative eukaryotic viruses with 30.2–92.7% amino acid identity to known viruses of 12 families (*Coronaviridae, Dicistroviridae, Hepeviridae, Iflaviridae, Nodaviridae, Parvoviridae, Picobirnaviridae, Picornaviridae, Permutotetraviridae, Polycipiviridae, Reoviridae* and *Totiviridae*) and to 7 currently unclassified viruses (Supplementary Table 1). Individual guano samples analyzed contained an average of 14.5 viruses (standard deviation 3.6) that varied in prevalence from 9 to 100% among samples and in abundance over approximately 4 orders of magnitude, with arthropod-infecting viruses generally most prevalent and most abundant, consistent with the insectivorous diet of *H. ruber* (Supplementary Fig. 2). Sequences corresponding to a novel betacoronavirus (*Coronaviridae: Betacoronavirus*) were present in 6 samples (55%) (Supplementary Fig. 2). Due to the public health significance of betacoronaviruses, we intensively sequenced this virus, Buhirugu virus 1 (BHRGV-1), using the sample with the most abundant reads (sample 9 in Supplementary Fig. 2) and succeeded in obtaining 15,433 bases of the orf1a/b polyprotein gene and 3,181 bases of the spike protein gene (GenBank OP199247). Phylogenetic analyses (Fig. 2) show BHRGV-1 to be a novel member of the subgenus *Hibecovirus*, approximately equidistant from Bat Hp-betacoronavirus and Zaria bat coronavirus[13,14]. BHRGV-1 and the other hibecoviruses form a well-supported clade most closely related to viruses of the separate subgenus *Sarbecovirus*, which contains SARS-CoV and SARS-CoV-2 (Fig. 2)[15].

To investigate the potential host range of BHRGV-1, we conducted predicted protein structure and in silico docking analysis of the BHRGV-1 spike protein (Supplementary Fig. 3) and the angiotensin II (ACE2) receptors of humans and the other mammals observed consuming bat guano (Supplementary Fig. 4). In cases where the ACE2 nucleotide sequence of a particular animal was not available, we used sequences from a closely related species. The Ramachandran scores for BGHRV-1 S and the various species of ACE2 within the energetically favored region of the protein ranged from 95 to 98% (Supplementary Table 3). Docking analyses of the BHRGV-1 spike protein indicate non-permissive binding interactions between BHRGV-1 S and ACE2 receptors in all sequences analyzed

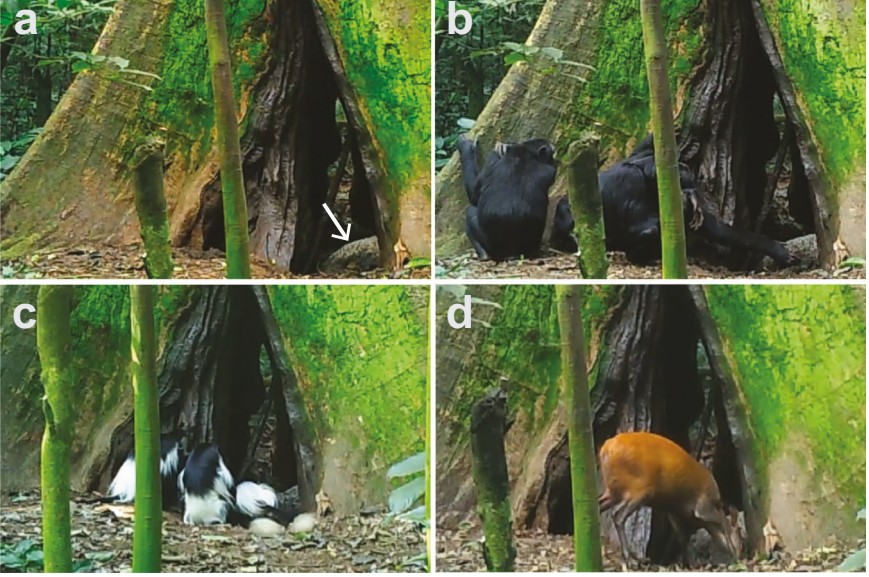

**Fig. 1 | Guano consumption by wildlife in Budongo Forest Reserve, Uganda.** Images of guano pile (**a**, arrow) and chimpanzees (**b**) black-and-white colobus (**c**), and red duiker (**d**) consuming bat guano were captured using trail cameras between July 5th and October 18th 2017 and between September 14th 2018 and April 28th 2019.

**Table 1 | Concentrations of minerals in bat guano, decaying wood, clay soil, termite mound soil, and normal fruit + leaf diet (mg/kg) in Budongo samples (means ± standard deviations)**

| Mineral element | Bat guano (n = 5) | Decaying Raphia farinifera wood (n = 11)[a] | Decaying Cleistopholis patens wood (n = 20)[a] | Clay soil (n = 10)[b,c] | Termite mound soil (n = 44)[b,c] | Normal diet (n = 24)[a] | Kruskal-Wallis[d] |
|---|---|---|---|---|---|---|---|
| Ca | 4280 ± 736 | 1563 ± 1176 | 5682 ± 6616 | 2381 ± 3003 | 3201 ± 3006 | 13315 ± 30648 | H = 30.72; $p < 0.0001$ |
| Fe | 1238 ± 24 | 128 ± 143 | 148 ± 160 | 8720 ± 3080 | 45728 ± 21250 | 649 ± 1310 | H = 92.26; $p < 0.0001$ |
| K | 23180 ± 1145 | 6650 ± 3366 | 10817 ± 17177 | 2528 ± 3613 | 908 ± 277 | 4074 ± 6485 | H = 50.91; $p < 0.0001$ |
| Mg | 5840 ± 1356 | 2430 ± 2262 | 2136 ± 2013 | 1012 ± 1165 | 669 ± 251 | 1557 ± 1272 | H = 30.83; $p < 0.0001$ |
| Mn | 706 ± 123 | 425 ± 521 | 50 ± 141 | 306 ± 252 | 954 ± 277 | 66 ± 69 | H = 89.00; $p < 0.0001$ |
| Na | 1822 ± 61 | 5038 ± 4118 | 1871 ± 3207 | 234 ± 228 | 5 ± 14 | 293 ± 507 | H = 96.45; $p < 0.0001$ |
| P | 26420 ± 1638 | 367 ± 323 | 1425 ± 2557 | 414 ± 534 | 664 ± 174 | 851 ± 964 | H = 26.99; $p < 0.0001$ |

[a]Data from Reynolds et al.[9], recalculated from original values.
[b]Data from Reynolds et al.[26]
[c]Data from Reynolds et al.[27]
[d]Kruskal-Wallis H statistic and associated p value for differences among food sources for each mineral element.

(Supplementary Table 4), implying that ACE2 may not be the *Hibecovirus* receptor. To investigate other potential receptors, we repeated this analysis on Aminopeptidase-N, Dipeptidyl peptidase 4, and CEACAM1[16], with similar results indicating non-permissive binding interactions between BHRGV-1 S and each of these molecules (Supplementary Table 4, Supplementary Figs. 5–7 and Supplementary Tables 2–4). Finally, we examined whether BHRGV-1 has a predicted hemagglutinin-esterase region, which would implicate use of O-acetylated sialic acids as a receptor[17], but we found no evidence of such a region by scanning for matches in the InterPro protein signature databases[18,19].

## Discussion

Minerals are essential for physiological functioning, growth, reproduction and immunity[20]. Minerals are also often limiting in the core diets of wild animals[21]. Some cave-dwelling invertebrates, fish and salamanders consume bat guano to obtain minerals in their nutrient-limited subterranean environments[22]. However, to our knowledge, guano ingestion by forest-dwelling mammals has not previously been reported. Bat guano also contains nutrients critical to plant growth, such as nitrogen, phosphate, and potassium, making guano an efficient and widely used fertilizer[23]. This may explain why people appear to have visited the same tree where we documented guano consumption by wildlife. We note that another betacoronavirus has been described in bat guano collected as fertilizer in Thailand[24], and that harvesting bat guano for this purpose is a widespread but under-appreciated practice that may increase pandemic risk[25].

Our results suggest that guano consumption by Budongo wildlife may be a behavioral adaptation to mineral scarcity. This inference is supported by a decades-long body of evidence showing that wildlife in Budono have responded to the disappearance of *R. farinifera* by seeking alternative mineral sources[9,26,27]. The guano contained concentrations of potassium, magnesium, sodium and phosphorus equal to or in excess of concentrations in other dietary sources. Past studies have shown that consumption of alternative sources of minerals by Budongo chimpanzees began with the disappearance of *R. farinifera*[9,26,27]. Black-and-white colobus and duiker have not been as intensively studied in Budongo, so it is unknown whether guano consumption is also a new behavior for these animals. Black-and-white colobus frequently consume soil, clay, aquatic plants and even cement, demonstrating extreme dietary plasticity with respect to mineral acquisition[28].

Guano consumption also appears to expose wildlife to bat-associated viruses, to the extent that the sequences we obtained represent infectious viruses. BHRGV-1 is a member of the subgenus *Hibecovirus*, which contains viruses that primarily infect bats of the genus *Hipposideros* but have also been documented in the bat genera *Macronycteris*, *Nycteris*, and *Rhinolophus*[29], and *Hibecovirus* is sister taxon to the subgenus *Sarbecovirus*, which contains SARS-CoV and SARS-CoV-2. Predicted protein structure analysis of the BHRGV-1 S protein indicates the highest structural similarity to the SARS-CoV S protein (Supplementary Fig. 3 and Supplementary Table 2). However, the binding affinity of the BHRGV-1 S protein for all ACE2 proteins examined (including those of *H. armiger*, a close relative of *H. ruber*) is low (Supplementary Fig. 4 and Supplementary Tables 3 and 4), indicating that ACE2 is likely not a viable receptor for host cell entry in these mammals. We obtained similar results for the putative alternative receptors Aminopeptidase-N, Dipeptidyl peptidase 4, and CEACAM1[16] (Supplementary Figs. 5–7 and Supplementary Tables 3 and 4), and we found no evidence that BHRGV-1 has a predicted hemagglutinin-esterase region that might bind O-acetylated sialic acids, as has been shown for other coronaviruses[17]. If none of these molecules are, in fact, receptors for BHRGV-1 and other hibecoviruses, this would merit further study, especially for predicting the host range and zoonotic potential of coronaviruses in subgenera other than *Sarbecovirus*.

Coronavirus infections of wildlife have not been documented in Budongo to date. Examining local wildlife for evidence of BHRGV-1 or similar viruses in feces could yield additional information about the breadth of species that might have been exposed to bat-borne coronaviruses.

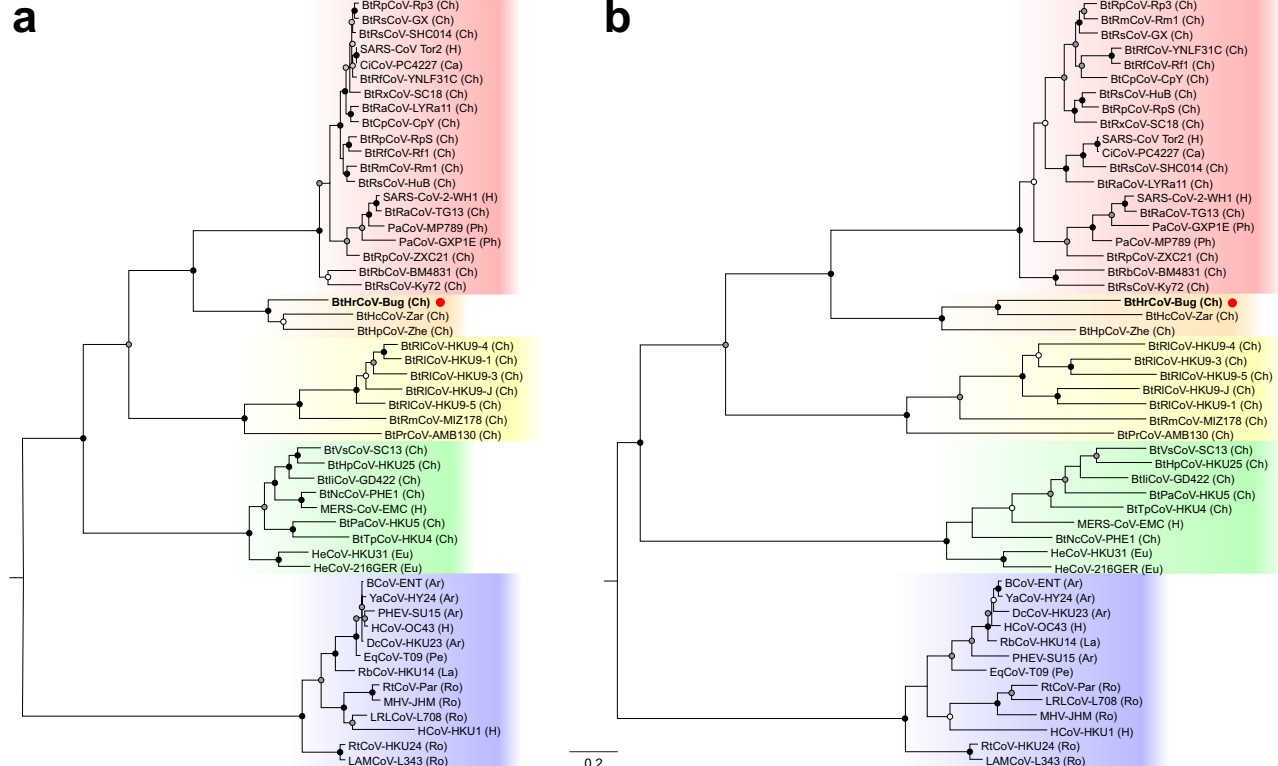

**Fig. 2 | Maximum likelihood phylogenetic trees of betacoronaviruses.** Trees of BHRGV-1 (bold, red dot) and representative betacoronaviruses based on the viral RNA-dependent RNA polymerase (**A**) and spike protein (**B**) genes were constructed from codon-based nucleotide alignments (15,816 and 5334 positions, respectively). Taxon labels are abbreviations (See Table S1 for full descriptions) with viruses from humans (H) or from hosts of other mammalian orders (Ar = Artiodactyla; Ca = Carnivora; Ch = Chiroptera; Eu = Eulipotyphla; La = Lagomorpha; Pe = Perissodactyla; Ph = Pholidota; Ro = Rodentia) in parentheses. Circles on nodes indicate bootstrap values based on 1,000 replicates (black = 100%; gray = 75%-99%; white = 50%-74%); only values ≥ 50% are shown. Colored groupings correspond to *Beta-coronavirus* subgenera (top to bottom): *Sarbecovirus*, *Hibecovirus*, *Nobecovirus*, *Merbecovirus*, and *Embecovirus*. Trees were midpoint rooted, yielding root placement the same as that of the International Committee on the Taxonomy of Viruses *Betacoronavirus* clade within the *Coronaviridae*[15]. Scale bar indicates nucleotide substitutions per site.

However, multiple outbreaks of respiratory disease in the chimpanzees of Budongo have been observed, with the causes remaining undiagnosed. Respiratory disease outbreaks in other chimpanzee populations in Uganda have resulted from cross-species transmission of viruses from humans[30,31], and human betacoronavirus OC43 can infect wild chimpanzees and cause clinical disease[32].

Intriguingly, chimpanzees, black-and-white colobus, and red duiker have all been implicated in ebolavirus outbreaks in Central and West Africa[1,33,34]. The natural history of the ebolaviruses is poorly understood, but multi-host models of sylvatic ebolavirus transmission posit that outbreaks occur when primates and ungulates become infected by bats and serve as amplifying hosts[2,3,35]. Similarly, many bat-borne coronaviruses have emerged in humans after transmission through intermediary hosts[4,5]. Mechanisms of virus transmission from bats to other wildlife in nature remain poorly understood, although consumption of fruit contaminated by bats[35,36] and contact with viruses shed into the environment[37,38] have been hypothesized. Our data suggest another plausible ecological mechanism for exposure of wildlife to bat-associated viruses: consumption of bat guano as a source of dietary minerals. Tropical forest plants and soils are mineral-poor[39]. Depletion of primary sources of minerals such as *R. farinifera* could create conditions that favor guano consumption as a "fallback" mineral source.

Infectious disease emergence is often attributed to drivers such as land conversion, hunting, urbanization, climate change, and agricultural intensification[40], but the ecological mechanisms whereby these drivers lead to cross-species pathogen transmission remain imprecisely understood. Our results provide an illustration of how these mechanisms might follow elaborate causal chains. In Budongo, international demand for tobacco caused local selective deforestation and loss of a primary source of dietary minerals, which led to fallback consumption of guano by wildlife and exposure of wildlife to bat-associated viruses, including a congeneric of the pandemic SARS coronaviruses. Mathematical tools for representing causal chains and causal networks are becoming widespread in epidemiology[41] and might prove useful for assessing how environmental and social drivers ultimately lead to zoonotic transmission. This understanding, in turn, could lead to improved precision in the application of tools for preventing pandemics. For example, compared to the costs of a pandemic, the costs of offering local farmers substitutes to *R. farinifera* for making strings to dry tobacco leaves would likely be trivial[6,11]. In general, we suggest that understanding causal chains and identifying their "breakable" links holds promise for illuminating disease ecology and improving zoonoses prevention.

## Methods
### Field studies
The study took place in the Budongo Forest Reserve, Uganda. We first observed chimpanzees in the habituated Waibira community[42,43] feeding on bat guano in a hollow tree on the 25th of June, 2017, even though chimpanzees had regularly been observed since 2011. On the 5th of July 2017, we installed a trail camera (Bushnell Trophy Cam, model 119,774) with the following default settings: 10 s video length and interval, auto sensor level, low night vision shutter, and 24 h camera mode. We mounted the camera to a tree 6 m from the tree atop the guano at a height of 1 m. Image capture occurred between the 6th of July and the 18th of October 2017 and again between the 14th of September 2018 and the 28th of April 2019. We

analyzed the resulting 14,567 10 s video recordings (40.46 camera-hours in total) for the presence and number of animals feeding on the guano. We defined an "instance" as a set of sequential video recordings of animals of a given species feeding on the guano, separated from the previous instance by at least 30 min during which no individuals of the same species were recorded.

We collected guano samples from this hollow tree monthly from 13th September 2018 to 29th April 2019 (Supplementary Fig. 2) and divided them for mineral content analysis and virus identification. We oven-dried samples for mineral content analysis (approximately 50 g) and stored and shipped them to the USA at ambient temperature. We placed samples for molecular analysis (approximately 0.9 ml) in 1.8 ml sterile cryovials, mixed them thoroughly with an equal volume of RNAlater Stabilization Solution (Thermo Fisher, Waltham, MA, USA), and stored them cold in the field and in liquid nitrogen prior to and during shipment to the USA.

All animal use was strictly non-invasive and observational. The study protocol was reviewed and approved by the Uganda Wildlife Authority and the Uganda National Council for Science and Technology, and was in compliance with the guidelines of the Animal Welfare and Ethical Review Body of the University of Stirling and all applicable regulations governing the protection of animals and research. We have complied with all relevant ethical regulations for animal use. The species of animals were *Pan troglodytes schweinfurthii*, *Hipposideros ruber*, *Colobus guereza occidentalis*, and *Cephalophus natalensis*, and all were wild-type and of undetermined sex and age.

### Dietary mineral analyses

Prior to analysis, we inactivated samples with ultraviolet radiation and oven-dried them at 105 °C for 48 h. We then digested samples using the MARS 6 Microwave Digestion System (CEM Corporation, Matthews, NC, USA) and analyzed them for Ca, P, Mg, K, Na, Fe, Zn, Cu, Mn, and Mo on an iCAP 6300 inductively coupled plasma radial spectrometer (Thermo Fisher, Waltham, MA, USA)[44].

### Metagenomic analyses

We processed guano samples for virus identification using metagenomic methods[31,45]. Briefly, we added 200 µl of guano+RNAlater to 800 µl of Hanks' balanced salt solution and homogenized them in PowerBead Tubes (Qiagen, Hilden, Germany) containing 2.38 mm metal beads. We then treated the homogenate with nucleases to reduce unencapsidated nucleic acids[46]. We used the QIAmp MinElute Virus Spin Kit (Qiagen, Hilden, Germany) to isolate total nucleic acids, and we and converted RNA to double-stranded cDNA using the SuperScript double-stranded cDNA Synthesis Kit (Invitrogen, Carlsbad, CA, USA). We cleaned cDNA using Agencourt AmpureXP beads (Beckman Coulter, Brea, CA, USA) and synthesized DNA libraries using the Nextera XT DNA sample preparation kit (Illumina, San Diego, CA, USA). We sequenced libraries on an Illumina MiSeq instrument using 600 cycle v3 MiSeq Reagent Kits. We trimmed resulting sequences at a Phred quality score <30, discarded reads <50 bp, and removed sequences matching host genomes and known contaminants. We thereby sequenced samples to a mean depth after quality and length trimming and host genome subtraction of 1.9 M reads (standard error 0.2 M reads), ranging from 1.4 to 3.5 M reads per sample.

### Data processing, bioinformatics, statistics, and reproducibility

We subjected sequence reads to de novo assembly using SPAdes 3.13.0[47], discarded resulting contiguous sequences (contigs) <500 nucleotides, and used cd-hit[48] to remove redundant contigs (90% similarity threshold). We compared remaining contigs to custom databases of representative virus protein sequences and to the NCBI non-redundant protein sequence database using blastx[49]. We ran and analyzed blank samples in parallel to ensure that cross contamination had not occurred.

To investigate BHRGV-1 in greater detail, we queried the initial 15,433 bp contig representing this virus against the National Center for Biotechnology Information (NCBI) nucleotide database using blastn in

BLAST + [50,51]. Based on this analysis, we chose two reference sequences for downstream comparisons: bat Hp-betacoronavirus/Zhejiang2013 (NCBI accession ID NC_025217.1) with 74.5% nucleotide sequence identity to BHRGV-1 and Zaria bat coronavirus strain ZBCoV (Genbank: HQ166910.1) with 76.4% nucleotide sequence identity to BHRGV-1. We then mapped sequence reads from the sample containing the highest concentration of this virus (sample 9, Fig. S2) against each reference separately using CLC Genomics Workbench (Qiagen, Hilden, Germany), specifying a minimum length fraction of 0.5 and a minimum similarity fraction of 0.8. We extended mapped regions in CLC Genomics Workbench using iterative mapping of previously unmapped reads to consensus sequences extracted from each prior iteration. We collected reads thus identified, assembled them de novo, and aligned the resulting contigs to both references to create a genome scaffold (each reference was useful for assembling different regions of the BHRGV-1 genome). We then mapped contigs and reads that did not form contigs against the genome scaffold to create a final draft 32,594 bp sequence of the BHRGV-1 ORF1ab polyprotein and spike protein (S) genes. We constructed phylogenetic trees of the BHRGV-1 ORF1ab polyprotein and spike protein (S) genes using PhyML[52] with smart model selection[53] (GTR + I models selected in both cases) and 1000 bootstrap replicates to assess statistical confidence in clades.

We modeled 3D-structures of the receptor binding domain (RBD) of BHRGV-1 S and the putative receptor proteins ACE2, Aminopeptidase-N, Dipeptidyl peptidase 4, and CEACAM1[16] from selected animal species using Modeler 10.2[54]. In cases for which a particular species lacked an available representative sequence, we chose the closest phylogenetic relative of that species for which a representative sequence was available: *Hipposideros armiger* (great roundleaf bat) in place of *Hipposideros ruber* (Noack's roundleaf bat); *Capra hircus* (goat) in place of *Cephalophus natalensis* (red duiker); and *Colobus angolensis palliates* (Angolan black-and-white colobus) in place of *Colobus guereza occidentalis* (black-and-white colobus) (Table S2). We modeled the BHRGV-1 S protein using the well characterized SARS-CoV spike fusion protein (PDB ID: 2BEZ) as a homologous structural template. Similarly, we used the human ACE2 (PDB ID: 1R42), Aminopeptidase-N (PDB ID: 5LHD), Dipeptidyl peptidase 4 (PDB ID: 2QT9), and CEACAM1 (PDB ID: 4QXW) proteins as homologous structural templates in the other species analyzed (Table S2). We evaluated the quality of the resulting models using the GA341 score[55], DOPE (Discrete Optimized Protein Energy) method scores[56], and the SWISS-MODEL structure assessment server[57]. We then refined structures with the lowest DOPE scores using molecular dynamics (MD) simulations and further analyzed them for quality using Ramachandran Plot and MolProbity in SWISS-MODEL (49) (Table S3).

We used HDOCK[58] to model putative receptor/BGHRV-1 binding complexes. HDOCK maps receptor and ligand protein molecules onto grids, then "docks" two molecules using a hierarchical approach based on fast Fourier transformation. To minimize bias, we applied the template-free docking method with structures generated by Modeler 10.2. We optimized the final docked protein complexes using the AMBER99SB-ILDN force field in GROMACS[59]. Specifically, docked complexes were immersed in a truncated octahedron box of TIP3P water molecules. The solvated box was further neutralized with Na + or Cl − counter ions using the tleap program. We used Particle Mesh Ewald (PME) to calculate long-range electrostatic interactions, with a cut-off distance for long-range van der Waals (VDW) energy term of 12.0 Å, and the system minimized without restraints. We applied 2500 cycles of steepest descent minimization followed by 5000 cycles of conjugate gradient minimization. We initiated MD simulations by heating each system in the NVT ensemble from 0 to 300 K for 50 ps using a Langevin thermostat with a coupling coefficient of 1.0/ps and a force constant of 2.0 kcal/mol·Å2 on the complex. We ran the MD simulation for 100 ns at a constant temperature of 300 K in the NPT ensemble with periodic boundary conditions for each system. During the MD procedure, we applied the SHAKE algorithm was to all covalent bonds involving hydrogen atoms, with a time step of 2 fs. We calculated free energies of binding for all simulated docked structures using the molecular mechanics Poisson

Boltzmann surface area (MM-PBSA) tool in GROMACS 2022[60] (Table S4). To examine BGHRV-1 for a predicted hemagglutinin-esterase region, we searched for matches in the InterPro protein signature databases using InterProScan 5.65-97.0[18,19].

## Data availability

All raw sequence reads were deposited in the NIH National Center for Biotechnology Information (NCBI) Sequence Read Achieve under BioProject PRJNA1087330 (accession numbers SAMN40440184-SAMN40440195). All assembled virus genome sequences were deposited in NCBI GenBank under accession numbers OP199247 and OP834146-OP834171.

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

## Acknowledgements

We are grateful to the Uganda Wildlife Authority and the Uganda National Council for Science and Technology for kindly granting permission to conduct field studies. We also thank the staff and management of Budongo Conservation Field Station for logistic support, and in particular Jacob Ariyo, Vicent Kiiza, Stephen Mugisha and Charles Rabu for assisting with field data collection. This work was supported in part through the European Research Council, grant agreement number 679787 ((to C. C.) and 802179) (to C.H.), the Royal Zoological Society of Scotland (to V.R.), the Armed Forces Health Surveillance Division (AFHSD), Global Emerging Infections Surveillance (GEIS) Branch, ProMIS ID P0167_22_NM (to K.A.B-L.), Navy WUN A1417 (to K.A.B-L.), and the University of Wisconsin-Madison John D. MacArthur Research Chair (to T.L.G.). The views expressed in this article are those of the authors and do not necessarily reflect the official policy or position of the Department of Defense, Department of the Navy, nor the U.S. Government. Several of the authors are U.S. Government employees. This work was prepared as part of their official duties. Title 17 U.S.C. § 105 provides that 'Copyright protection under this title is not available for any work of the United States Government.' Title 17 U.S.C. §101 defines a U.S. Government work as a work prepared by a military service member or employee of the U.S. Government as part of that person's official duties.

## Author contributions

P.F., C.A., and T.L.G. contributed to the study conception and design. P.F., C.A., W.J.A., V.R., C.H., R. K., G.M., K.Z. and C.C. contributed to sample and data collection in the field. P.F., J.M.R., G.K.R., R.Z.C. A.J.B., K.B.L. and T.L.G. conducted laboratory analyses. G.K.R., R.Z.C., A.J.B., K.B.L. and T.L.G. performed statistical analyses and interpretations. P.F. and T.L.G. wrote the initial manuscript, and all authors read, edited and approved the final manuscript.

## Competing interests

The authors declare no competing interests.

## Ethics approval

The study protocol was reviewed and approved by the Uganda Wildlife Authority and the Uganda National Council for Science and Technology and was in compliance with the guidelines of the Animal Welfare and Ethical Review Body of the University of Stirling and all applicable regulations governing the protection of animals and research. We have complied with all relevant ethical regulations for animal use.
