## [Peer review file · Communications Biology]

Reviewers' comments:

Reviewer #1 (Remarks to the Author):

The authors analyzed over 800 video-recorded instances of forest species feeding on guano in Budongo Forest, Uganda—a first record of forest-dwelling animals consuming guano. They hypothesize that this behavior is correlated with the disappearance of a mineral-rich palm tree species, as guano was found to be high in minerals. Furthermore, the guano was discovered to harbor a multitude of bat-borne viruses. In summary, this chain of events may contribute to zoonotic spillover. Additionally, a novel betacoronavirus is described, including the modeling of epidemiologically relevant elements such as the spike protein to ACE2 receptors, which did not appear to match

It has long been hypothesized that periods of food scarcity would increase competition and inter-specific interactions, thereby potentially causing the emergence of zoonotic spillover chains. Similarly, viral shedding in the environment has been proposed as an important mechanism for spillover generation.

Here, the authors could have gone a little deep into the literature regarding viral shedding from bats and other species into the environment which is said to promote spillovers. Here are some suggested example publications which are welcome to look into:

Becker, D.J., Eby, P., Madden, W., Peel, A.J. and Plowright, R.K., 2023. Ecological conditions predict the intensity of Hendra virus excretion over space and time from bat reservoir hosts. *Ecology Letters*, 26(1), pp.23-36.

Forbes, K.M., Sironen, T. and Plyusnin, A., 2018. Hantavirus maintenance and transmission in reservoir host populations. *Current opinion in virology*, 28, pp.1-6.

The novelty in this present manuscript lies in the video evidence collected over a significant period of time, providing rare and robust support for long-standing hypotheses. Additionally, the authors take an extra step by demonstrating the persistence of bat-sourced viruses in the guano, whether active or not. While the paper does not serve as a definitive proof of finding local zoonotic viruses like Ebola in the guano or documenting actual spillover events, it serves as a solid foundation for unraveling the chain of events that lead to spillovers, based on environmental factors and human impact. Especially since existing studies primarily either generate these hypotheses which are most commonly tested through modeling approaches.

While the manuscript may not be explicitly hypothesis-driven, the authors should be commended for their extensive efforts in collecting a substantial body of evidence for this new behavior, rather than merely publishing a few initial observations.

The animals were observed consuming "bat guano and guano- and urine-contaminated clay and water from under a large, hollow tree." It raises the question of whether the animals are specifically targeting the guano or if its consumption is merely a byproduct of consuming the clay. Please clarify how you knew these behaviours were distinct.

I assume that the greyish object in the Figure and videos in the main text represents the guano pile. It may be beneficial for the authors to enhance the clarity by adding a white triangle or arrow to

indicate the exact location of the guano pile within the image. Additionally, considering the clear depiction of the guano pile in the Supplementary figure with an arrow next to the tree, it could be advantageous to move this figure to the main text for better visibility and understanding.

L81 “repeatedly consuming apparently large volumes of guano” -> not sure what apparently large means, this seems speculative in the absence of data on amounts eaten.

L113 “Metagenomic analysis of the bat guano revealed 27 eukaryotic viruses (Supplementary Table 1)” → Please summarize here briefly the families these viruses belong to so we’re not relegated to the supplementary as only source of information

L124 add full stop at end of sentence

L129 add closing parenthesis after respectively

L153 add a space after full stop

The Supplementary figures with the modeled spike protein and ACE binding sites are hard to read - they are very pixelated / bad quality. If these were screengrabs, a better way to publish them would be to print the screen as pdf and convert to a high resolution png before inserting it into the document.

Reviewer #2 (Remarks to the Author):

This manuscript provides an interesting opportunistic exploration into plausible guano-mediated viral transmission between bats and non-human primates in the context of rapid environmental change. The manuscript is well-written and makes a sound case for the progression of steps taken to investigate the consequences of behavioral change observed in wildlife, especially primates. Please see line edits below, particularly regarding the phylogenetic analyses and interpretations.

As one major comment, the manuscript would be strengthened if the authors were able to show that BHRGV-1 can also be recovered from guano-foraging wildlife (e.g., in feces), as that would demonstrate capacity for viral transmission. Currently, I agree that the authors demonstrate the capacity for exposure to occur (viruses in guano that wildlife are consuming), but is there evidence that the viruses found in guano are also detectable in the community feeding on guano (either from samples collected during this expedition or from the authors’ past primate fecal sampling efforts)?

L35: The statement about ebolavirus hosts doesn’t seem particularly important for the abstract, given that no filoviruses were identified in guano samples (and bats have yet to be fully implicated as competent hosts for ebolaviruses specifically).

L71: As those authors note later, they have identified a novel hibecovirus, one of several subgenera in Betacoronavirus. This is indeed a sister clade to the sarbecoviruses (including SARS-CoV and SARS-CoV-2), but the mention of SARS-CoV-2 feels a bit forced here. The identified virus isn’t really closely related to SARS-CoV-2 and statements like the sentence here could even be mis-interpreted in negative ways that hamper bat conservation efforts. It would be more appropriate to state the result plainly, that the authors identify a novel betacoronavirus within the hibecovirus subgenus.

L136: It would help to provide the underlying evolutionary model selected by SMS in PhyML for the trees here. Have the authors considered rooting the tree with a relevant outgroup?

L137: See my earlier comment about noting SARS-CoV/SC2 connections. I think the text here is much more appropriate than in the Introduction, but a brief qualifier here could still emphasize that this virus is from an entirely distinct subgenus (e.g., “BHRGV-1 and the other hibecoviruses form a well-supported but entirely distinct clade adjacent to viruses of the subgenus Sarbecovirus”)

L182: See my earlier comment, but can relevant sequences be identified in any metagenomic data from the wildlife feces?

L193: I would suggest replacing “have not been determined” with “remain poorly understood”, as there are multiple examples of plausible transmission routes (e.g., co-roosting as in Lehmann et al. 2020).

L228: Were bat guano samples (e.g., as noted in L112) collected from this one roost? Different timepoints? How many samples total?

Reviewer #3 (Remarks to the Author):

The authors describe important and novel findings that contribute to our understanding of key interfaces that may facilitate spillover of viruses from bats to wildlife and/or humans. Overall the manuscript is well-written and appropriate for publication in *Communications Biology*. However, I think the causative link between wildlife consumption of guano and mineral scarcity is not well-supported by the data presented as the authors have not shown that this behaviour is directly correlated to mineral scarcity. The data presented here describe the behaviour and variation in mineral content, but the data do not provide statistical support variation of the behaviour across species, time, or space relating to mineral scarcity and, therefore, the data cannot be used to support the hypothesis that mineral deficiencies are driving changes in this behaviour. I would appreciate if the authors could soften conclusions keeping this distinction in mind.

The authors tested only ACE2 which is the receptor for one alphacoronavirus and several betacoronavirus, which the virus described here phylogenetically clusters with the hibecoviruses for which no receptor has been characterized. The authors have provided no justification for the selection of just ACE2 to test using relatively easily using their *in silico* approach. The paper would be enhanced by additional *in silico* docking analyses with other known coronavirus receptors including APN, DPP4, and CECAM1. It would also be helpful to know if the authors detected a predicted hemagglutinin-esterase region which would implicate use of O-acetylated sialic acids as a receptor which is the case for hCoV-HKU1 and hCoV-OC43.

The authors must include an accession number for their Buhirugu virus in the manuscript (which I believe is OP199247?) and the raw data should be submitted to a public repository such as NCBI SRA prior to publication.

Minor comments below:

Supplemental: Figures here are low-resolution and difficult to interpret

Line 35-36, this phrasing is a bit misleading as NHPs and duiker are not considered likely maintenance hosts of ebolaviruses.

Line 124: add period at the end of the sentence and accession number to public repository (e.g. NCBI)

Line 129: missing end parenthesis

Figure 2: I cannot distinguish between grey and white nodes in this tree, please place colour grouping boxes on background layer.

Line 140: Please replace “molecular modelling” with predicted protein structure and in silico docking analysis.

Line 153: add space before “Minerals”

Line 194: Please phrase as “an additional plausible ecological mechanism” as this paper does not supplant consumption of contaminated fruit as a route of exposure which is well-supported by data.

“Selective deforestation and exposure of African wildlife to bat-borne viruses”

For publication in *Communications Biology*

Point-by-Point Responses to Reviewer Comments

Reviewer #1

Comment 1: Here, the authors could have gone a little deep into the literature regarding viral shedding from bats and other species into the environment which is said to promote spillovers. Here are some suggested example publications which are welcome to look into:

- Becker, D.J., Eby, P., Madden, W., Peel, A.J. and Plowright, R.K., 2023. Ecological conditions predict the intensity of Hendra virus excretion over space and time from bat reservoir hosts. *Ecology Letters*, 26(1), pp.23-36.
- Forbes, K.M., Sironen, T. and Plyusnin, A., 2018. Hantavirus maintenance and transmission in reservoir host populations. *Current opinion in virology*, 28, pp.1-6.

Response: Thanks, those are indeed relevant publications. We have expanded the text on viral shedding from bats and other species into the environment and its relationship to spillover events, and we now cite these publications (Line 195).

Comment 2: While the manuscript may not be explicitly hypothesis-driven, the authors should be commended for their extensive efforts in collecting a substantial body of evidence for this new behavior, rather than merely publishing a few initial observations.

Response: Thank you! We very much appreciate the encouraging comments!

Comment 3: The animals were observed consuming "bat guano and guano- and urine-contaminated clay and water from under a large, hollow tree." It raises the question of whether the animals are specifically targeting the guano or if its consumption is merely a byproduct of consuming the clay. Please clarify how you knew these behaviours were distinct.

Response: We thank the reviewer for the carefully considered comment. As described above in our response to Editor comments 1, the videos clearly show that the animals were most often consuming the guano itself, directly selecting pieces of guano (and not clay) then eating them. We have therefore modified this paragraph to clarify that guano consumption was not a byproduct of clay or water consumption, including the evidence supporting that conclusion (Lines 82-85).

Comment 4: I assume that the greyish object in the Figure and videos in the main text represents the guano pile. It may be beneficial for the authors to enhance the clarity by adding a white triangle or arrow to indicate the exact location of the guano pile within the image. Additionally, considering the clear depiction of the guano pile in the Supplementary figure with an arrow next to the tree, it could be advantageous to move this figure to the main text for better visibility and understanding.

Response: This is a good point. We have added a panel to Figure 1 with a new image clearly showing the guano pile by itself, with an arrow indicating its precise location.

Comment 5: L81 “repeatedly consuming apparently large volumes of guano” -> not sure what apparently large means, this seems speculative in the absence of data on amounts eaten.

Response: Indeed, we are not sure either! We have removed the phrase “apparently large volumes of” (Line 81).

Comment 6: L113 “Metagenomic analysis of the bat guano revealed 27 eukaryotic viruses (Supplementary Table 1)” → Please summarize here briefly the families these viruses belong to so we’re not relegated to the supplementary as only source of information.

Response: Thanks, this is a good idea. One should not have to dig deep into supplemental data for such information! We have added a summary of the viral families, as requested (Lines 105-108).

Comment 7: L124 add full stop at end of sentence.

Response: Done; thanks for catching the typo (Line 108).

Comment 8: L129 add closing parenthesis after respectively.

Response: Fixed; thanks again.

Comment 9: L153 add a space after full stop.

Response: Added; thanks again.

Comment 10: The Supplementary figures with the modeled spike protein and ACE binding sites are hard to read -they are very pixelated / bad quality. If these were screengrabs, a better way to publish them would be to print the screen as pdf and convert to a high resolution png before inserting it into the document.

Response: We’re not sure why the figures appeared this way, but thank you for bringing it to our attention. We have performed more receptor modeling (see comments to R2, below) and all the modeling figures are new. We have ensured that they now included in their original high-resolution format.

Reviewer #2

Comment 1: As one major comment, the manuscript would be strengthened if the authors were able to show that BHRGV-1 can also be recovered from guano-foraging wildlife (e.g., in feces), as that would demonstrate capacity for viral transmission. Currently, I agree that the authors demonstrate the capacity for exposure to occur (viruses in guano that wildlife are consuming), but is there evidence that the viruses found in guano are also detectable in the community feeding on guano (either from samples collected during this expedition or from the authors’ past primate fecal sampling efforts)?

Response: We agree with the Reviewer that testing fecal samples obtained from animals consuming guano for the presence of BHRGV-1 would strengthen arguments about the

potential for viral transmission. We have not collected such fecal samples, nor do we have such data from previous studies (see response to Editor's comment 2, above). However, our intent is to submit grant proposals for just this sort of study, once this manuscript is published. We have therefore added a statement about the potential utility of such studies to the Discussion (Lines 181-183).

Comment 2: L35: *The statement about ebolavirus hosts doesn't seem particularly important for the abstract, given that no filoviruses were identified in guano samples (and bats have yet to be fully implicated as competent hosts for ebolaviruses specifically).*

Response: Agreed, thanks. We've removed the statement about ebolaviruses as suggested (Line 35).

Comment 3: L71: *As those authors note later, they have identified a novel hibecovirus, one of several subgenera in Betacoronavirus. This is indeed a sister clade to the sarbecoviruses (including SARS-CoV and SARS-CoV-2), but the mention of SARS-CoV-2 feels a bit forced here. The identified virus isn't really closely related to SARS-CoV-2 and statements like the sentence here could even be mis-interpreted in negative ways that hamper bat conservation efforts. It would be more appropriate to state the result plainly, that the authors identify a novel betacoronavirus within the hibecovirus subgenus.*

Response: Yes, that's true, and we certainly don't want to hamper bat conservation efforts! As described in our response to Editor comment 3, we have modified this sentence as the reviewer suggests (Line 72).

Comment 4: L136: *It would help to provide the underlying evolutionary model selected by SMS in PhyML for the trees here. Have the authors considered rooting the tree with a relevant outgroup?*

Response: Good point! We've provided the underlying evolutionary model (Line 280). We did try rooting with an outgroup, but the alignment for doing so forces the omission of over 50% of phylogenetically informative sites within the betacoronaviruses (for spike). We therefore selected midpoint rooting but checked this rooting against the most recent ICTV phylogeny of the *Coronaviridae* (Woo et al. 2023, J Gen Virol Apr;104(4). doi: 10.1099/jgv.0.001843). Fortunately, our rooting is identical to the ICTV rooting of the betacoronavirus clade within the larger coronavirus phylogeny. We have added a statement in the Results to this effect (Lines 540-542).

Comment 5: *See my earlier comment about noting SARS-CoV/SC2 connections. I think the text here is much more appropriate than in the Introduction, but a brief qualifier here could still emphasize that this virus is from an entirely distinct subgenus (e.g., "BHRGV-1 and the other hibecoviruses form a well-supported but entirely distinct clade adjacent to viruses of the subgenus Sarbecovirus").*

Response: We understand the reviewer's point and think it's a very good suggestion. We've added the qualifier "separate" to emphasize the distinctness of the two clades (Line 121). Thank you!

Comment 6: L182: *See my earlier comment, but can relevant sequences be identified in any metagenomic data from the wildlife feces?*

Response: We appreciate the reviewer's curiosity, and we share it! We compared our sequence to all sequences available in GenBank and the SRA and have not identified any such sequences, and we don't at present have additional samples to analyze. As per the reviewer's first comment, we have added a statement about the utility of such studies to the Discussion (Lines 181-183).

Comment 7: L193: *I would suggest replacing “have not been determined” with “remain poorly understood”, as there are multiple examples of plausible transmission routes (e.g., co-roosting as in Lehmann et al. 2020).*

Response: Agreed, that wording is much better. We've made the change (Line 194).

Comment 8: L228: *Were bat guano samples (e.g., as noted in L112) collected from this one roost? Different timepoints? How many samples total?*

Response: Good question. We provide complete information on samples in the Supplementary Materials, but we've now added a summary of this information to the main text, as the reviewer suggests (Lines 229-230).

Reviewer #3

Comment 1: *However, I think the causative link between wildlife consumption of guano and mineral scarcity is not well-supported by the data presented as the authors have not shown that this behaviour is directly correlated to mineral scarcity. The data presented here describe the behaviour and variation in mineral content, but the data do not provide statistical support variation of the behaviour across species, time, or space relating to mineral scarcity and, therefore, the data cannot be used to support the hypothesis that mineral deficiencies are driving changes in this behaviour. I would appreciate if the authors could soften conclusions keeping this distinction in mind.*

Response: Thank you, yes, perhaps we were a bit over-exuberant. We agree and have softened the language and conclusions regarding the causal link between selective deforestation and bat guano consumption throughout the manuscript. We have also clarified that these inferences are based on a strong body of prior research documenting similar behaviours involving other food items such as clay, termite mounds, and decaying trees. Thus, the link between guano consumption and mineral scarcity is not a “stand-alone” finding, but rather adds to a multi-decade body of evidence (see response to Editor comment 1, above; Lines 67-68 and 82-85 and 152-154).

Comment 2: *The authors tested only ACE2 which is the receptor for one alphacoronavirus and several betacoronavirus, which the virus described here phylogenetically clusters with the hibeoviruses for which no receptor has been characterized. The authors have provided no justification for the selection of just ACE2 to test using relatively easily using their in silico approach. The paper would be enhanced by additional in silico docking analyses with other known coronavirus receptors including APN, DPP4, and CECAM1.*

Response: The reviewer's point is well taken. We have conducted additional in silico docking analyses for APN, DPP4, and CECAM1 (this effort accounts for our long delay in resubmitting this manuscript). We have added a supplementary tables and figures

(Supplementary Table 4, Supplementary Figures 5-7 and Supplementary Tables 2-4) presenting the results and have added text to the Methods (Lines 132-138), Results (Lines 123-138) and Discussion (Lines 167-179) sections. Although this was a difficult endeavor, we appreciate the suggestion as it clearly improves the manuscript.

Comment 3: *It would also be helpful to know if the authors detected a predicted hemagglutinin-esterase region which would implicate use of O-acetylated sialic acids as a receptor which is the case for hCoV-HKU1 and hCoV-OC43.*

Response: Thanks, that's a very good suggestion too! We have conducted this analysis and describe it in the Results section (Lines 136-138), and we discuss the new results in the Discussion (Lines 174-176) and describe the methods fully in the Methods (Lines 317-319), respectively.

Comment 4: *The authors must include an accession number for their Buhirugu virus in the manuscript (which I believe is OP199247?) and the raw data should be submitted to a public repository such as NCBI SRA prior to publication.*

Response: The reviewer is correct, and the data have indeed been submitted to NCBI. The accession number is OP199247 (Line 118), and we will alert NCBI to modify the record to reflect the proper publication and links to other data sources once the paper is accepted.

Comment 5: *Supplemental: Figures here are low-resolution and difficult to interpret.*

Response: Fixed; please see response to R1, comment 9, above.

Comment 6: *Line 35-36, this phrasing is a bit misleading as NHPs and duiker are not considered likely maintenance hosts of ebolaviruses.*

Response: True, they are suspected to be intermediary hosts between the reservoir (unknown) and humans. We have deleted this text (Line 35).

Comment 7: *Line 124: add period at the end of the sentence and accession number to public repository (e.g. NCBI).*

Response: Thanks, added.

Comment 8: *Line 129: missing end parenthesis.*

Response: Thanks again, fixed.

Comment 9: *Figure 2: I cannot distinguish between grey and white nodes in this tree, please place colour grouping boxes on background layer.*

Response: Thanks! Yes, that was hard to see. We've moved the color shading to the background layer and have darkened the shading of the grey nodes so the grey and white nodes are now easy to distinguish.

Comment 10: *Line 140: Please replace "molecular modelling" with predicted protein structure and in silico docking analysis.*

Response: This wording is much more precise, thanks. We have made the change (Lines 123-124 and 167).

Comment 11: Line 153: add space before “Minerals.”

Response: Thanks for catching this typo. We’ve made the change.

Comment 12: Line 194: Please phrase as “an additional plausible ecological mechanism” as this paper does not supplant consumption of contaminated fruit as a route of exposure which is well-supported by data.

Response: This is also a good suggestion. We have changed the phrase (Line 196). Thank you!

REVIEWERS' COMMENTS:

Reviewer #2 (Remarks to the Author):

I thank the authors for addressing my previous concerns!

Reviewer #3 (Remarks to the Author):

The authors have sufficiently revised the manuscript following reviewer comments. This is an exciting study which is appropriate for publication in communications biology.